# Electrochemical and In Vitro Biological Evaluation of Bio-Active Coatings Deposited by Magnetron Sputtering onto Biocompatible Mg-0.8Ca Alloy

**DOI:** 10.3390/ma15093100

**Published:** 2022-04-25

**Authors:** Ana-Iulia Bița, Iulian Antoniac, Marian Miculescu, George E. Stan, Lucia Leonat, Aurora Antoniac, Bujor Constantin, Norin Forna

**Affiliations:** 1Department of Metallic Materials Science and Physical Metallurgy, Materials Science and Engineering Faculty, University Politehnica of Bucharest, 060042 București, Romania; ana.iulia.bita@upb.ro (A.-I.B.); iulian.antoniac@upb.ro (I.A.); aurora.antoniac@upb.ro (A.A.); 2National Institute of Materials Physics, 405A Atomistilor Street, 077125 Măgurele, Romania; lucia.leonat@infim.ro; 3Department of Quality Engineering and Industrial Technologies, Industrial Engineering and Robotics Faculty, University Politehnica of Bucharest, 060042 București, Romania; cbujor2006@yahoo.com; 4Department of Orthopedics and Traumatology, University of Medicine and Pharmacy Gr.T.Popa Iasi, 16 University Str., 700115 Iasi, Romania; norin.forna@umfiasi.ro

**Keywords:** bio-active coatings, hydroxyapatite, bio-glass, Mg-0.8Ca alloy, magnetron sputtering

## Abstract

The use of resorbable magnesium alloys in the design of implants represents a new direction in the healthcare domain. Two main research avenues are currently explored for developing or improving metallic biomaterials: (i) increase of their corrosion resistance by designed compositional and structural modifications, and (ii) functionalization of their surfaces by coating with ceramic or polymeric layers. The main objective of this work was to comparatively assess bio-functional coatings (i.e., highly-crystallized hydroxyapatite and silica-rich glass) deposited by radio-frequency magnetron sputtering (RF-MS) on a biodegradable Mg-0.8Ca alloy (0.8 wt.% of Ca). After probing their morphology (by scanning electron microscopy) and structure (by Fourier transform infrared spectroscopy and grazing incidence X-ray diffraction), the corrosion resistance of the RF-MS coated Mg-0.8Ca substrates was electrochemically tested (in synthetic biological media with different degrees of biomimicry), and their cytocompatibility was assessed in osteoblast and fibroblast cell cultures. By collective assessment, the most promising performances, in terms of mass loss (~7% after 12 days), hydrogen release rate (~6 mL/cm^2^ after 12 days), electrochemical corrosion parameters and cytocompatibility, were obtained for the crystalline HA coating.

## 1. Introduction

The classic metallic biomaterials used for the fabrication of implants for internal fixation of bone fractures (e.g., austenitic stainless steels; titanium super-alloys) are bioinert and must be removed by subsequent surgical procedures. Such procedures increase the medical expenses and, to a greater extent, the patient’s morbidity [1,2]. Thus, the major advantage conferred by the potential use of resorbable metal biomaterials in the execution of such type of implants emerged as a necessity.

Although pure magnesium (Mg) is a biodegradable material, its corrosion rate is generally too fast to be used for medical applications. Moreover, Mg is not homogeneous, owing to its accentuated tendency for localized corrosion. Another problem is the release of hydrogen during its corrosion. If the hydrogen leaching is too fast, then the volume of gas is too large and cannot be absorbed, leading to the accumulation of gas in tissues. Furthermore, the subsequent change of pH in the immediate vicinity of the implant is another cause for serious concern. For the improvement of the corrosion resistance of Mg and its alloys, two main methods are prominently implemented: (i) adjusting the composition and microstructure of the material by selecting suitable alloying elements [3,4]; and (ii) performing surface treatments/applying protective ceramic or polymeric coatings [5,6,7,8,9,10].

The most frequent Mg alloying elements are Al, Zn, Mn, Li, Zr, Ca, and rare-earth elements [11,12]. Amongst them, Ca is known, if used in low amounts, to increase the mechanical resistance by solid solution and precipitation hardening [13]. To a large extent, Ca can also act as a grain refining agent and contribute to the grain boundary strengthening. In Mg-Ca binary alloys, CaMg_2_ Laves-type intermetallic phases accompany the major α-Mg constituent [14]. The use of a high amount of Ca (>1 wt.%) can cause problems with hot casting [15]. In terms of health, Ca is an essential element for the physiology and biochemistry of cells, being also an important component of hard tissues. The presence of Ca in Mg alloys was found to be beneficial for a faster healing of bone tissue [16,17]. However, the inadequate mechanical properties and low corrosion resistance of Mg-Ca alloys are considered, rightfully so, as major limitations [15,18]. These disadvantages can be mitigated by controlling the Ca content, as well as by thermo-mechanical processing treatments.

The solubility limit of Ca in Mg is 1.34 wt.%. When Ca content increases, a higher amount of CaMg_2_ is obtained, with higher granulation, precipitating at the grain boundaries and having effects on the alteration of both the mechanical properties and the corrosion resistance [19,20]. Therefore, it can be considered that the positive effect of grain refining in Mg-Ca alloys occurs at a Ca content of maximum 1 wt.%. Although electrochemically the intermetallic CaMg_2_ compound is more active than pure Mg, the in vitro and the in vivo studies have shown that small Ca additions (up to 1 wt.%) lead to the reduction of grain size in the casting structure and the increase of the corrosion resistance [21,22].

The corrosion phenomena of the binary Mg-Ca alloys have gathered significant interest [23]. Li et al. published the first complex study dedicated to these materials, exploring compositional series of binary Mg-Ca alloys [23]. They showed that a higher content of CaMg_2_ phase led to better corrosion resistance. Furthermore, mechanical properties such as yield strength, ultimate tensile strength, and elongation were found to be conveniently engineered by controlling the Ca concentration. Noteworthily, the Mg-1Ca alloy (1 wt.% Ca), in particular, induced the new bone formation three months after implantation in New Zealand rabbits. Zreiqat et al. demonstrated that Mg ions can facilitate the adhesion of human bone-derived cells, owing to the increase of the α5β1- and β1-integrin receptors levels [24]. Hydrogen releases was detected only in the initial phase of implantation [25]. This can be explained by an accelerated corrosion taking place on the surface of the Mg alloys and to the secondary acidosis resulting from the metabolic and resorbable processes occurring immediately after surgery [26]. Without special surface treatment, the hydrogen molecules are self-absorbed, since the traces of H_2_ were reported to disappear after several weeks [23].

All this in vitro and in vivo evidence advocates for the great promise of biodegradable Mg-Ca alloys for orthopaedic applications, but also indicates that several further improvements are required. For instance, a synchronization of the biodegradation rate of the Mg alloy-based implants with the rate of new bone formation needs to be attained for highly-performant Mg-Ca alloys. Such hopes can be tentatively achieved by the surface modification of the biodegradable alloys with highly biocompatible and adherent coatings of a ceramic, polymeric, or composite nature, eliciting more controlled degradation speeds [27,28]. An ideal coating should provide protection to the Mg-based alloy implant in period immediately subsequent to surgery, while later, when the formation and growth of new bony tissue occurs, it should incrementally degrade.

Legitimately, the obvious candidate coating materials are represented by the those capable of osseointegration (i.e., generation of a direct structural and functional bond between the host living bone and the surface of the implant [29]), namely calcium phosphates—prominently hydroxyapatite (HA) [30,31] and bioactive glass (BG) [32,33]. Furthermore, carbonated HAs are known to more closely resemble the composition of the mineral component of the bone [31], and also to elicit superior biological performance with respect to pure HA [34,35] and satisfactory mechanical properties in both porous scaffold [31] and coating [34] form. On the other part, silica-based BGs present the highest bioactivity index amongst known biomaterials [36]. Furthermore, recent evidences [32,37,38,39,40] have indicated that the reduction or even the elimination of alkali oxides from the composition of BG favours a higher durability to fast solubilisation, reduces their coefficient of thermal expansion, and mitigates the possible events of burst pH increase (owed to the leaching of high doses of alkali ions).

Joining the worldwide efforts to identify viable coating solutions for the functionalization of biodegradable Mg-alloys, our group advanced as working variants highly crystalline carbonated HA and alkali-reduced silica-rich bioactive glass thin films deposited by the industrial ready (i.e., easy to scale-up) radio-frequency magnetron sputtering technique (RF-MS) [41]. Their morpho-structural features were only preliminarily tested, whilst their adherence to the Mg-Ca-based substrates was found to be of ~56 (for HA films) and ~33 (for BG films) MPa [41], thus superior to the minimum value of 15 MPa imposed the ISO 13779-2:2008 standard. These primary results endorsed both the further exploration of their physical–chemical features and the testing of their in vitro biological and corrosion performances using protocols with an increased degree of biomimicry.

So far, such tests were mostly performed in various simplistic, inorganic-based media (e.g., Tris-HCl, Earle’s or Hank’s balanced salt solutions, phosphate buffered saline, Kokubo’s simulated body fluid (SBF)) [42,43,44,45], whilst the true intercellular fluid has a far more complex composition, with amino acids, proteins, enzymes, and other organic moieties coming into play [46,47,48]. The presence of the organic component is known to influence the corrosion process (either anodic, cathodic, or both) [49,50,51] and thereby, their presence may be another factor influencing the in vivo biocorrosion of the Mg-Ca alloys [52]. Several studies have shown that the addition of foetal serum of bovine origin in the simulated synthetic media environments has changed the corrosion behaviour of Mg alloys compared to the one recorded in purely inorganic media [53,54,55,56,57].

Thereby, on the path towards feasible temporary implants based on Mg alloys, the (i) rational selection of the system components (e.g., alloy composition, type of surface modification, thermal processing) should satisfy three main requirements—biocompatibility, corrosion rate (inhibited or transiently delayed), and conservation of adequate mechanical properties during degradation), should be merged with the (ii) implementation of exigent biological and corrosion testing protocols, fit to refine only the best material/technical solutions. Biocompatibility and corrosion resistance must be considered as central requirements for the biodegradable implant, so that its presence and degradation will not affect the biological functions of the body [58,59,60]. Possibilities to improve mechanical properties can be subsequently searched by thermo-mechanical processing.

In congruence with this view, the main objective of this work was to comparatively evaluate the structural, surface energy, corrosion (in both mainly inorganic—SBF, and complex organic-inorganic—Dulbecco’s Modified Eagle Medium—simulated body media) and cytocompatibility properties of hydroxyapatite and silica-rich glass thin layers deposited by RF-MS onto a Mg-0.8Ca-type (0.8 wt.% of Ca) biodegradable magnesium alloy.

## 2. Materials and Methods

### 2.1. Fabrication of the Mg-0.8Ca Alloy

The Mg-0.8Ca alloy was obtained in an electric tilting furnace equipped with crucibles (type CT-AL-1,1) and KANTAL resistors, using graphite crucibles and an instant thermocouple. The raw materials were constituted by Mg in compact form and Ca granules (stored in argon). The pure Mg was preheated to 150 °C and then melted at 680 °C in a sulphur dioxide atmosphere. At this step, the Ca granules were added to the melt, and after mixing, the filler was poured when the melt reached 620 °C. The casting was carried out in multiple bar cast iron shell moulds, preheated to 250 °C. After solidification and cooling in air, the bars having a diameter of 10 mm were deburred and cleaned. Square- or disc-shaped samples with size/diameter of 12 or 15 mm, and thickness of 2 mm were cut by sawing. Subsequently, they were polished with metallographic SiC paper (down to a grit size of P1000), and then degreased (in acetone) and cleaned (in absolute ethanol) by ultrasonication.

### 2.2. Deposition of Coatings by RF-MS

Cathode targets were manufactured by pressing the source materials into copper dishes—powders of pure synthetic HA (Sigma Aldrich, St. Louis, MO, USA) and an in-house prepared (by melt-quenching) SiO_2_-CaO-P_2_O_5_-Na_2_O-MgO silica-based glass [41]. The HA2- and BG2-type thin films were deposited onto single crystal <100> Si wafers (for deposition rate determinations) and Mg-0.8Ca substrates (for functional characterizations) by RF-MS using a Vacma UVN-75R1 (Kazan, Soviet Union) equipped with a 1.78 MHz RF generator and applying the work protocol described in our previous work [41]. Briefly, the HA2 and BG2 layers, were prepared using a target-to-substrate separation distance of 35 mm, argon (purity 6N, Linde, Dublin, Ireland) working atmosphere, gas flow of 39 sccm, and sputtering pressures of 0.3 (HA2) and 0.4 Pa (BG2). First depositions were carried out for 60 min on Si wafers to determine by spectroscopic ellipsometry and cross-sectional scanning electron microscopy (SEM) the thicknesses of the films, and thus, the deposition rate of each type of material. Subsequently, using the inferred deposition rates of ~8 (HA2) and 5.5 (BG2) nm/min, films with a thickness of ~1 µm were deposited onto the Mg-0.8Ca substrates.

The amorphous as-deposited HA films were crystallized post-deposition in air at 500 °C/1 h (a temperature found optimal, as it enables the complete crystallization of HA film without the excessive oxidation of the Mg-based alloy substrate). To test the ability of the HA layer to play a dual role—(i) bio-functional and (ii) buffer against the oxidation of the Mg-0.8Ca substrate during the post-deposition crystallization heat-treatment in air, the annealing was performed simultaneously for batches of uncoated Mg-0.8Ca substrates as well. The composition of the BG film was intentionally enriched in silica by fixing fused silica coupons on top of the racetrack of the magnetron cathode target, following a procedure proven effective in previous works [46,61]. An increase in silica of the glass films enables the increase of the network connectivity, having the decrease of solubility and the improvement of mechanical properties as consequences [46,61].

### 2.3. Methods of Characterization

#### 2.3.1. Physical–Chemical Analysis

The cross-sectional and surface morphology of the bare and RF-MS coated Mg-0.8Ca alloy specimens was analysed by field emission high-resolution SEM (FE-HRSEM) by employing a Carl Zeiss Gemini 500 apparatus (Carl Zeiss Company, Oberkochen, Germany). The micrographs were recorded under high vacuum (~2–7 × 10^−4^ Pa) at an electron high tension voltage of 5 kV and a working distance of 7–8 mm.

The surface wettability was assessed by water contact angle (CA) measurements, performed with a Krüss Drop Shape Analyzer (model DSA100). The experiments were carried out at a temperature of 20 ± 1 °C and a humidity of 45 ± 5%. The images were captured 5 s after the deposition of the water droplet. The recorded data was processed with the ImageJ software (National Institutes of Health, Bethesda, MD, USA).

The structure of the Mg-0.8Ca alloy was investigated by X-ray diffraction (XRD) in Bragg Brentano geometry, with a Bruker D8 Advance apparatus (with CuK_α_ radiation and an 1D LynxEye detector). The diffractogram was recorded in the 2θ range of 20–90° with a step size of 0.02°. The phase composition and the mean crystallite size of the alloy’s constituents were determined by Rietveld whole powder pattern fitting [62] using the Topas (version 3.0) software.

The crystalline status/phase composition of the films was inferred by grazing incidence X-ray diffraction (GIXRD) using a Rigaku SmartLab 3 kW system (Rigaku Corporation, Tokyo, Japan) with CuK_α_ radiation (λ = 1.5418 Å). The incidence angle was set at 2° and the diffraction patterns were acquired in the 2θ range of 20–50° with a step size of 0.02°.

The chemical structure of the samples was investigated by Fourier-transform infrared (FTIR) spectroscopy in attenuated total reflectance (ATR) mode, using a Jasco 6800-FV-BB spectrometer (Jasco Corporation, Tokyo, Japan) equipped with ATR PRO670H attachment with bulk diamond crystal. The FTIR-ATR spectroscopy measurements were performed under vacuum within the wave numbers range of 4000–100 cm^−^^1^, at a resolution of 4 cm^−^^1^.

#### 2.3.2. Functional Performance of the RF-MS Coatings

Functional performance of the ceramic coatings was evaluated by corrosion tests. The determination of generalized corrosion resistance was performed comparatively by immersion tests, determining the weight loss, the amount of released hydrogen, and the electrochemical behaviour. Two testing media were used: the (a) Kokubo’s simulated body fluid (SBF) prepared according to the recipe given in [63] and the (b) Dulbecco’s Modified Eagle Medium (DMEM, Gibco, Evansville, IN, USA) frequently used for cell culturing. For comparison, the composition of the two types of testing media are given in Table 1.

##### Weight Loss and Hydrogen Release Determinations

The experiments were performed by immersing simple and RF-MS-coated Mg-0.8Ca alloy specimens (in form of square coupons with side of 12 mm and height of 2 mm) in a volume of 45 mL of SBF or DMEM, at a pH of 7.4 and a temperature of 37 ± 1° C, for 3, 5, 7, 10, and 12 days. The experiments were performed in quintuplicate.

At the end of each testing time period, the samples were removed from the solution, cleaned in distilled water, dried in a desiccator and then passed through a chromic acid solution to remove degradation products. Subsequently, the samples were washed and dried. The samples were weighed initially (before immersion) and after removal of degradation products, and the weight loss and the degradation/corrosion rate were calculated.

##### Electrochemical Behaviour

To determine the corrosion resistance, a corrosion cell was used, consisting of a saturated calomel electrode (SCE)—the reference electrode, a platinum electrode—the recording electrode, and the working electrode which consisted of the investigated samples. The samples were mounted in a Teflon support so that the surface subjected to corrosion had an equal area of 100 mm^2^. During the corrosion tests, the electrochemical cell was introduced in a Faraday cage to eliminate interferences due to the electromagnetic fields.

The tests were performed in both the SBF and DMEM media, at the physiological temperature of 37 ± 0.5 °C, using a heating and recirculation bath (JeioTech, model CW-05G, Seoul, Korea).

##### Morpho-Compositional Evaluation by SEM-EDS

Subsequent to the immersion and electrochemical tests, the surface of all samples was analysed by SEM coupled with energy dispersive X-ray spectrometry (EDS) to evaluate the surface morphological and compositional modifications.

#### 2.3.3. Cytocompatibility Assays

Two types of cell lines were used: human osteosarcoma (SaOs-2, ATCC HTB-85) and fibroblasts (Hs27, ATCC CRL1634). DMEM low glucose (Gibco, Evansville, IN, USA) supplemented with 10% foetal bovine serum (FBS) was used for the osteoblast cells culturing. DMEM high glucose (Gibco, Evansville, IN, USA) supplemented with 5% FBS was used for the fibroblast cells culturing.

The simple and HA2- and BG2-coated Mg-0.8Ca specimens were incubated in each of the two media for 24 and 72 h under physiological conditions (5% CO_2_, 21% O_2_, humidity 95%, 37 °C). After 24 and 72 h, the samples were removed and the medium was further used in the framework of the cell viability/proliferation tests. The cell culture tests were performed with the extracted medium for 1, 2, 3, and 7 days.

Before initiating the tests, the cells were cultured in plates and grown until they reached a confluence of 80%. The control of the experiments was constituted of medium incubated with plain HDPE cell culture surface. The cells’ viability/proliferation was inferred by an Alamar Blue test (CellTiter-Blue^®^, Promega, Madison, WI, USA), following the manufacturer protocol. Briefly, the following steps were carried out:

(1)Cells were seeded in 96-well plates at a density of 2000 cells/100 μL DMEM-FBS culture medium and left to adhere and develop for 24 h in the incubator.(2)The cell culture medium was removed, cells being further treated with 100 µL of sample extracts and control. The plate was incubated for 1, 2, 3 and 7 days.(3)At every time frame the medium from each well was carefully removed and changed with 130 µL of the CellTiter-Blue^®^ solution (10% *v*/*v* of DMEM). After 2.5 h of incubation, 100 µL CellTiter-Blue^®^ solution with cells was transferred to a matte white plate and the fluorescence was measured using a plate reader.

All tests were performed in triplicate, and the results were presented as arithmetic means ± standard deviations.

## 3. Results and Discussion

### 3.1. Surface Analysis: Morphology and Wettability

The HA2 and BG2 films deposited onto Si substrates, using shorter sputtering times, were cleaved and their morphology was investigated by cross-sectional SEM (Figure 1). Thicknesses of ~480 and ~330 nm were inferred for the HA2 (Figure 1a) and BG2 (Figure 1b) coatings, confirming the deposition rates of ca. 8 and 5.5 nm [41], respectively, previously estimated on the basis of spectroscopic ellipsometry measurements. Furthermore, it was observed that both type of films were conformal (having a definite, smooth interface with the substrate), well-adhered, and compact.

FE-HRSEM images characteristic for the surface of the uncoated and the HA2- and BG2-coated Mg-0.8Ca specimens are presented in Figure 2. The surface of the as-fabricated Mg-alloy substrate (Figure 2a,b) presented fine grooves, a result of the cutting (from the ingot) and SiC-polishing procedures; however, such a rough morphology could provide adequate in-situ fixation of the implant. After the 500 °C/1 h annealing in air, the morphology of the Mg-0.8Ca surface changed, becoming covered with a blanket of tightly-packed grains with sizes situated in the range of 80–170 nm (Figure 2c,d). This suggested that new compounds have formed on the metallic surface.

The deposition of the HA2- (known to have a Ca/P molar ratio of ~1.7 [41]) and BG2-type (with oxide composition in mol%: SiO_2_—58, CaO—18, P_2_O_5_—2, Na_2_O—3, MgO—19 [41]) coatings changed the substrate morphology, which became constituted of an envelope with a fine microstructure, uniformly covering the topographic surface faults of the substrate (Figure 2e–h). Both type of RF-MS layers featured agglomerates, consequence of the Volmer-Weber growth mode (based on the nucleation of islands, coalescence, and formation of a continuous compact coating) and prominent self-shadowing phenomena [64,65]. Polyhedral-shaped grains (with sizes in the range of ~150–350 nm), protruding from the film matrix, were evidenced in the case of HA2 films only (Figure 2e,f), suggesting the success of the post-deposition crystallization heat-treatment performed at 500 °C/1 h.

The bare Mg-0.8Ca alloy had a CA with water of ~75°. The coating with BG2- and HA2-type coatings determined the increase of the CA to values of ~83° and ~105°, respectively, thus shifting the surface character more and more towards the hydrophobicity (i.e., CA > 90°).

### 3.2. Structural Investigation

The XRD pattern (collected in Bragg–Brentano configuration) of the Mg-0.8Ca alloy is shown in Figure 3. In this geometry, the penetration depth of the X-rays in Mg is the range 50–80 µm, thus resulting in a relevant sampling analysis depth. The XRD results indicated that the Mg-0.8Ca alloy consisted of a major hexagonal (space group P63/mmc) Mg phase (ICDD-PDF4: 04-004-8745) and a minor hexagonal (space group P63/mmc) CaMg_2_ compound (ICDD-PDF4: 00-013-0450). The CaMg_2_ share resulted from Rietveld analysis was of ~1 wt.%. The average crystallite size of the Mg and CaMg_2_ phases were estimated at ~100 and ~80–90 nm, respectively. The Mg matrix elicited a slight preferential orientation of crystallites with the (001) planes parallel to the sample surface. This was inferred on the basis of the March-Dollase coefficient, obtained by Rietveld fitting, which had a value of 0.87 (thus smaller than 1, which characterizes an isotropic sample). The CaMg_2_ was found to be isotropic.

The XRD patterns (collected in grazing incidence (GI) configuration, α = 2°) of the bare and RF-MS coated Mg-0.8Ca specimens are comparatively presented in Figure 4a. In this geometry, the penetration depth of the X-rays in Mg is of ~11 µm, being thus more sensitive towards surface-formed compounds and also facilitating the structural analysis of thin films, since it determines a longer path of the incident beam in their reduced volume (with respect to the symmetric geometry). The GIXRD analysis (Figure 4a) indicated that the as-fabricated Mg-0.8Ca contained in its surface, besides the Mg and CaMg_2_ (unveiled by the Bragg–Brentano XRD measurements), two supplemental minor phases: magnesium oxide (ICDD-PDF-4: 01-077-8619) and magnesium peroxide (ICDD-PDF4: 00-019-0771). These could be the result of the alloy surface oxidation in ambient conditions. The annealing of the uncoated Mg-0.8Ca produced the formation on the surface of two new compounds (i.e., calcite—ICDD-PDF4: 04-007-0049 and calcium oxide—ICDD-PDF4: 04-010-5778), along with a pronounced increase in intensity of the magnesium peroxide compound, testimony of a strong surface oxidation during the annealing process. In good agreement with FE-HRSEM data, the XRD patterns confirmed the complete crystallization of the HA layer (whose diffraction maxima were seamlessly matched by a hexagonal hydroxyapatite phase, ICDD-PDF4: 00-009-0432) and the amorphous status of BG2 one (evidenced by the broad amorphous halo spanning in the 2θ region of ~15–45°, see Figure 4a-inset). Remarkably, in the case of HA2-coated samples (heat-treated at 500 °C/1 h in air), the diffraction peaks of the MgO and MgO_2_ phases did not experience a noteworthy change (with respect to the uncoated as-fabricated Mg-0.8Ca specimen). This indicated that the HA2 magnetron sputtered layer is compact (in good agreement with the cross-sectional SEM results—Figure 1a), and thus served as a buffer layer against substrate oxidation during the post-deposition crystallization annealing.

The FTIR spectra (Figure 4b) offered further structural insights. The 500 °C annealed Mg-0.8Ca specimens featured the juxtaposed contribution of IR bands specific to Mg–O and Ca–O bonds, peaking in the spectral region 750–350 cm^−1^ [66,67,68,69,70,71,72]. The formation of CaCO_3_ of the surface of this type of samples, as indicated by the GIXRD data (Figure 4a), is confirmed by the presence of the characteristic IR bands of carbonate groups in calcite: symmetric deformation (ν_4_) at ~691 cm^−1^, asymmetric deformation (ν_2_) at ~884 cm^−1^, and asymmetric stretching (ν_3_) at ~1409 and ~1499 cm^−1^ [59,60]. The lattice mode bands of MgO, MgO_2_, CaO, and CaCO_3_ appear in the far-IR region of 300–150 cm^−1^ [68].

The HA2 film featured sharp and split IR absorption bands (indicative of a well-crystallized material), representative for the vibrational modes of orthophosphate (i.e., triply degenerated asymmetric stretching (ν_3_) at ~1089, 1063, and ~1018 cm^−1^, non-degenerated symmetric stretching (ν_1_) at ~963 cm^−1^, triply degenerated bending (ν_4_) at ~603 and 563 cm^−1^, and doubly degenerated bending (ν_2_) at ~474 cm^−1^), hydroxyl (i.e., stretching at ~3572 cm^−1^ (not shown), librational at ~629 cm^−1^, and translational at ~332 cm^−1^), and sub-lattice 2[(CaII)_3_-(OH)] and libratory motions of phosphate (~300–150 cm^−1^) in a hydroxyapatite compound [73,74]. Furthermore, the carbonated nature of the HA2 film was exposed by the presence of the symmetric (ν_2_) and asymmetric (ν_3_) stretching modes of carbonate [73,74], peaking at ~878 and 1421, 1485, and 1546 cm^−1^, respectively. The BG2 films elicited a FTIR spectrum with broad maxima, specific to an amorphous material. Vibrational modes specific to a silica-based glass were identified: i.e., stretching of the Si–O–Si bonds in all the silicate units at ~1185 (TO_3_, transverse-optical) and ~1062 (LO_3_, longitudinal-optical) cm^−1^, stretching of the Si–O^−^ bonds in Q^2^ and Q^3^ silicate tetrahedra (with one and two non-bridging oxygen atoms) at ~987 cm^−1^, bending of the Si–O–Si bridges at ~788 cm^−1^, and rocking motion of the Si–O–Si bridges at ~518 cm^−1^ [38,61,75].

### 3.3. Bio-Functional Properties Assessment

#### 3.3.1. Weight Loss

The mass evolution of the specimens, determined at different immersion (in DMEM and SBF testing media) time intervals (3–12 days), of the bare and HA2- and BG2-coated Mg-0.8Ca samples are shown in Figure 5a,b. The best results were obtained for the HA2-coated Mg-0.8Ca samples. After 12 days of immersion, the weight loss was found to be of only ~7% (in DMEM) and ~26% (in SBF) with respect to the bare (uncoated) Mg-0.8Ca specimens, which recorded weight losses of ~24% (in DMEM) and ~81% (in SBF). It is thus suggested that the HA2-type layer provided an effective protection for the Mg-0.8Ca substrate, slowing down its corrosion rate. For the BG2-coated samples, a very small weight loss was observed up the 3rd day of immersion (irrespective of type of testing medium) with respect to the HA2-coated samples. However, starting from the 7th day of soaking in the simulated body media, the mass loss was accelerated. A mass loss of ~12% (in DMEM) and of ~41% (in SBF) was recorded for BG2-coated specimens, thus being higher with respect to the HA2-functionalized ones, but lower compared to the uncoated Mg-0.8Ca alloy. If in the case of the RF-MS coated metallic substrates, the mass loss had a quasi-linear increase, in the case of the bare Mg-0.8Ca alloy, a plateau was reached, starting with the 7th day of immersion. This suggested that the Mg(OH)_2_ layer, known to form on the surface of the alloy in such conditions, began to become stable over time [9].

#### 3.3.2. Hydrogen Release

Overall, the hydrogen release rates, recorded for all types of samples, were lower in DMEM compared to SBF, being more marked for the non-functionalized Mg-0.8Ca material (Figure 6a,b). In the case of the uncoated Mg-0.8Ca samples, the hydrogen release rate measured in the SBF solution exceeded, after the first day of immersion, the rate of hydrogen adsorption that can be tolerated by the human body [41]. The hydrogen release rate of the bare Mg-0.8Ca increased progressively and rather linearly until the 12th day in SBF, whilst in DMEM a monotonous increase was noticed until the 7th day, followed by an increase in an exponential fashion up to the 12th day. In good agreement with the mass loss results (Figure 5a,b), the BG2-coated specimens presented quite low hydrogen release rates up to the 3rd day of immersion. Subsequently, the release rate increased abruptly until the end of the testing period for BG2, being higher than that of the HA2-coated samples starting with the 6th day. It is thus indicated that the corrosion rate increased significantly, and that the BG2 layer was strongly damaged owned to its higher solubility with respect to HA. This can result in the weakening of the coating adhesion to the substrate. The HA2-coated samples showed promising evolution in terms of hydrogen release rates, being markedly superior with respect to both the uncoated and BG2-coated specimens. The hydrogen release rate of the HA2 samples slowed down after the 9th day (216 h) of immersion. This might be determined by the concomitant formation of stable biomimetic apatite in-growths on the surface of these samples, providing additional protection against the corrosive environment.

#### 3.3.3. Electrochemical Behaviour

The superimposed Tafel curves of the bare and coated Mg-0.8Ca samples, immersed in the DMEM and SBF testing media, are presented in Figure 7a,b, respectively. The main parameters of the electrochemical corrosion processes, which took place in the DMEM and SBF testing media, are presented in Table 2 and Table 3, respectively. The values of the open circuit potential (E_oc_) allow us to establish the “noble” character of the investigated samples. The polarization resistance was determined following the ASTM G59-97 (2003) standard. The corrosion resistance of the samples was examined on the basis of several evaluation criteria: (i) a more electropositive corrosion potential (E_cor_); (ii) a lower corrosion current density (i_cor_); and (iii) a higher polarization resistance (R_p_) are indicative for an improved corrosion resistance.

The electrochemical measurements have shown that, when using DMEM as an electrolyte, all samples had a negative corrosion potential with values lower than −1 V. The HA2- and BG2-coated samples presented close (−1.5 V) and lower (−1.1 V) values, respectively, in terms of electropositivity, with respect to the bare Mg-0.8Ca substrate (−1.4 V). When using SBF as the electrolyte, it is observed that the more electropositive corrosion potentials are produced by the HA2- and BG2-coated samples (of −1.69 V and −1.75 V, respectively), thus eliciting a better corrosion behaviour with respect to the metallic substrate (−1.86 V). When tested in DMEM solution, the BG2-coated samples had the lowest current density (≈0.87 µA/cm^2^), followed by the HA2-coated ones (≈2.07 µA/cm^2^). In the case of tests performed in SBF, the lowest current density was obtained for the HA2 samples (≈62.2 µA/cm^2^). Both the HA2- and BG2-coated samples tested in DMEM presented higher R_p_ values with respect to the bare Mg-0.8Ca material. When using SBF as an electrolyte, the only type of coating which led to a noteworthy improvement in terms of R_p_ values was the HA2 one.

Overall, lower i_cor_ and higher R_p_ values were recorded for the samples tested in DMEM, thus indicating their diminished corrosion, in good agreement with the mass loss (Figure 5) and hydrogen release (Figure 6) results.

From the corrosive attack efficiency (P_e_) point of view, the following conclusions can be drawn: (a) when using DMEM as testing medium, values higher than 90% are obtained (i.e., ≈95.9% and 98.3% for the HA2- and BG2-coated samples, respectively); (b) when using SBF as testing medium, the HA2 coating provided the best protection (P_e_ ≈ 89.2%) against the corrosive attack for the Mg-0.8Ca alloy.

#### 3.3.4. SEM/EDS Analysis of Immersed Samples

The surface morphology (surveyed by SEM) and composition (investigated by EDS) of the samples immersed in DMEM and SBF are presented in Figure 8 and Figure 9, respectively. The SEM analyses revealed that the HA2 layer provided a better corrosion resistance with respect to the BG2 one, in both DMEM and SBF environments. The surface of the HA2-coated samples was more homogeneous, being covered by a thin layer composed of acicular fine crystals (more marked in the case of the SBF-tested specimens). This morphology is characteristic to the biomimetic apatite precipitated in simulated body media [46,63], and further warrants their better corrosion resistance. This was also supported by the modification of the intensity ratio of the Ca K and P K peaks in the EDS spectra (Figure 8 and Figure 9). After the corrosion tests, the cracked surface appearance of the BG2-coated specimens was rather similar to the one the bare Mg-0.8Ca alloy, irrespective of type of testing medium (Figure 8 and Figure 9). This suggested a pronounced, but not a complete dissolution of the BG2 film, since the EDS spectra still indicate the presence of the Si K peaks of too high intensity to be associated to remnant embedded particles resulting from the SiC polishing of the Mg-0.8Ca substrates. Therefore, the BG2-type coating will provide a lower corrosion protection to the Mg-0.8Ca alloy, with respect to the HA2 functionalization solution. The presence of intense Ca K and P K peaks for all samples tested in SBF, including the uncoated Mg-0.8Ca, hints towards the spontaneous precipitation of calcium phosphate-based compounds. This is known to occur at alkaline pH values, which can be easily reached by this type of purely inorganic medium [46]. On the other part, the overall reduced corrosion observed in DMEM medium can be linked to both its more potent pH buffer and to its organic component, since such moieties, found also in the real intercellular fluid, are known to be adsorbed on the surface of the tested samples, acting as a capping layer [46]. It should be noted that a consensus with respect to a proper in vitro testing medium for the corrosion resistance and mass loss (degradation) is yet to be reached, since evidence amongst various studies is sometimes conflicting. For instance, Walker et al. [76] indicated that the Earle’s Balanced Salt Solution (EBSS)—a collection of the electrolytes found in human plasma (in analogous amounts), similar to SBF, but supplemented with glucose—can be used with success in the framework of in vitro corrosion testing protocols as a suitable predictor of the in vivo corrosion (evaluated on Lewis rats). The addition of organic components to EBSS increased the in vitro corrosion rate of Mg-based alloys (including Mg-0.8Ca). In another study, Gao et al. [77] tested the in vitro electrochemical and degradation behaviour of calcium phosphate-coated Mg-alloy implants in SBF and compared the results with those of the in vivo assays performed in animal model (white rabbits). They noticed that the in vitro electrochemical tests in SBF can be considered as fast screening tools for the corrosion resistance, whilst the conventional in vitro immersion assays led to degradation rates 2–4 folds faster than the ones recorded in vivo. It is thus difficult to advance at this point a definite resolution, since numerous factors must be accounted for, including but not limited to type of pH buffer (and its ability to maintain the solution pH at the normal physiological values); medium composition, relevance, and stability, or correct homeostatic testing ambient. Systematic and insightful studies, performed on the same type of specimens under the same conditions, in the framework of round-robin interlaboratory tests, are needed.

#### 3.3.5. Cytocompatibility Assays

The osteoblast and fibroblast cells viability/proliferation results obtained in the extracts harvested for all the uncoated and coated Mg-0.8Ca samples, after 24 and 72 h, are presented as bar-charts in Figure 10a,b (for osteoblasts) and Figure 10c,d (for fibroblasts), respectively. In the case of the 24 h extracts, all tested samples recorded good and rather similar cytocompatibility, irrespective of cell phenotype, with the sole exception of the BG2 and Mg-0.8Ca extracts at 7 days of fibroblast culturing. In the case of the 72 h extracts, all samples recorded a marked decrease of both osteoblast and fibroblast cell viability/proliferation at the 7-day timeframe. Interestingly, the HA2 and BG2 72 h sample extracts delivered analogous biological performance in osteoblast cell culture, whilst in the case of fibroblasts, a superior cytocompatibility could be emphasized. In the 8-days period from seeding, the cells can multiply about 32 times (osteoblast doubling time ~36 h). In this time frame, the cell density increases while resources become scarce, leading the cell resistance to diminish in the face of stress factors such as the leached ions present in the extracts. The low proliferation viability recorded in the case of fibroblasts incubated for 7 days with the 72 h extracts can be attributed to the lower doubling time of this cell phenotype (~26–28 h), which increased the cell stress.

## 4. Conclusions

Crystalline carbonated hydroxyapatite (HA) and silica-rich glass (BG) layers were successfully deposited by magnetron sputtering onto biodegradable Mg-0.8Ca alloys. Their good conformity was probed by SEM analyses, whilst XRD results have indicated that the HA layer is able to mitigate the metallic substrate oxidation during the post-deposition crystallization annealing in air.

The electrochemical and in vitro biological evaluation of the bio-active coatings deposited onto Mg-0.8Ca alloys was further comparatively assessed. It was noticed that the BG layer deteriorates faster than the HA one, providing inferior protection against the corrosive environment. Overall, the most promising results, in terms of mass loss, hydrogen release rate, electrochemical corrosion parameters (irrespective of type of testing medium—SBF or DMEM), and cell viability alike, were obtained for the crystalline HA-coated sample. The comparative multi-parametric analysis recommended this delineated functionalization solution for further insightful exploration. For instance, the degradation of HA-coated magnesium-based alloys might be tuned by controlling the surface wettability and thickness of the deposited layer.

## Figures and Tables

**Figure 1 materials-15-03100-f001:**
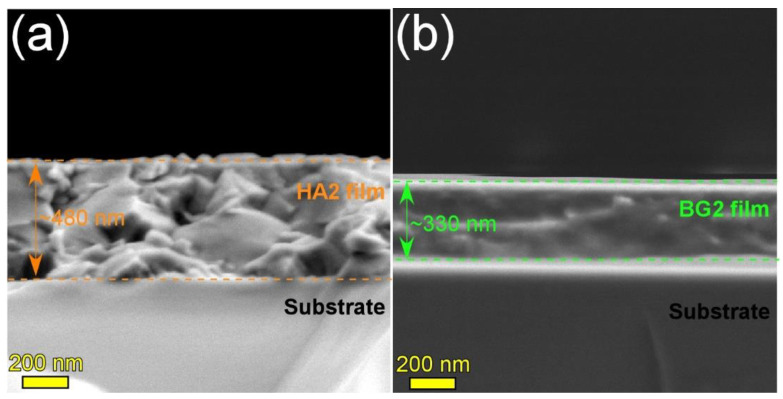
Cross-sectional SEM images of the (**a**) HA2 (post-deposition heat-treated at 500 °C/1 h in air) and (**b**) BG2 (as-sputtered) films deposited onto Si substrates.

**Figure 2 materials-15-03100-f002:**
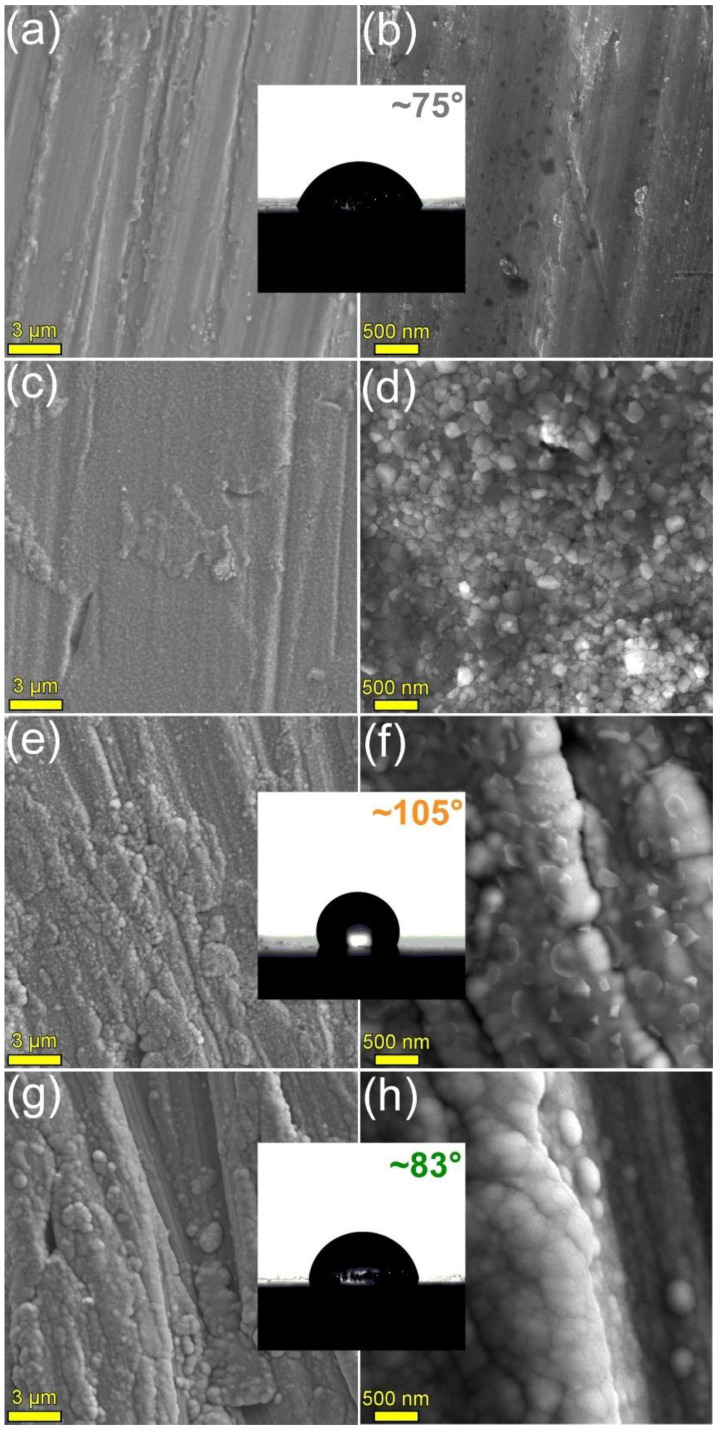
Characteristic FE-HRSEM images for the surface of the (**a**,**b**) bare as-fabricated, (**c**,**d**) bare 500 °C annealed in air, (**e**,**f**) HA2 (post-deposition heat-treated at 500 °C/1 h in air), and (**g**,**h**) BG2 (as-sputtered) RF-MS functionalized Mg-0.8Ca substrates. Insets: Typical CA images recorded for the surface of the bare and HA- and BG2-coated Mg-0.8Ca specimens.

**Figure 3 materials-15-03100-f003:**
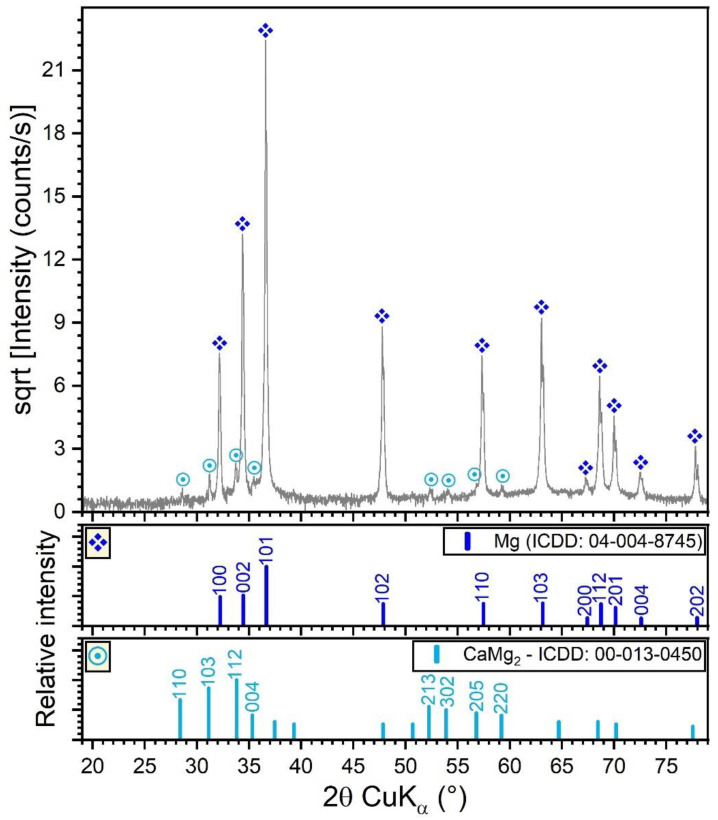
XRD diagram (collected in Bragg–Brentano geometry) of the Mg-0.8Ca alloy.

**Figure 4 materials-15-03100-f004:**
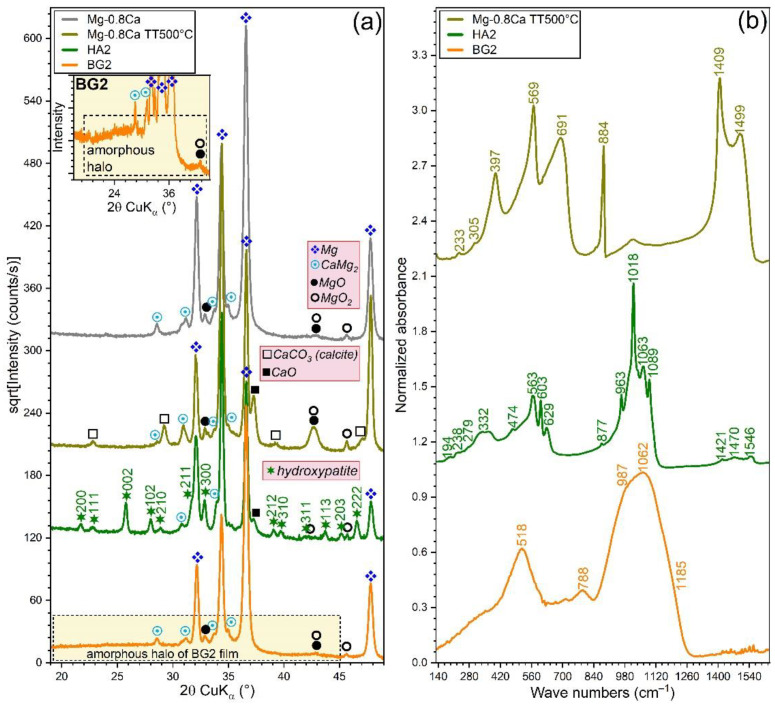
(**a**) XRD patterns (recorded in grazing incidence mode, α = 2°) and (**b**) FTIR-ATR spectra of the bare as-fabricated and 500 °C-annealed Mg-0.8Ca substrates and of the HA2- (post-deposition heat-treated at 500 °C/1 h in air) and BG2- (as-sputtered) RF-MS functionalized Mg-0.8Ca substrates. Inset in (**a**): Zoomed region (rescaled on the intensity *Y*-axis) allowing to visualize the amorphous hump, specific to the BG2-type film.

**Figure 5 materials-15-03100-f005:**
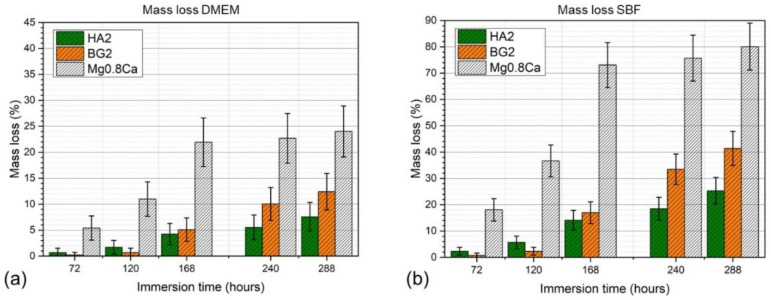
Mass loss time evolution in (**a**) DMEM and (**b**) SBF testing media of the bare and HA2- and BG2-coated Mg-0.8Ca specimens.

**Figure 6 materials-15-03100-f006:**
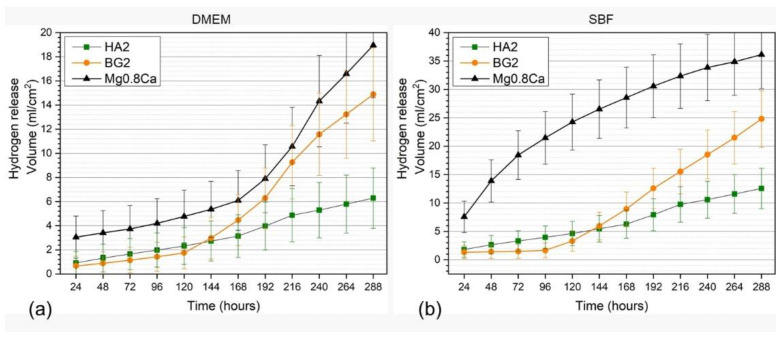
Hydrogen release evolution in (**a**) DMEM and (**b**) SBF testing media of the bare and HA2- and BG2-coated Mg-0.8Ca specimens.

**Figure 7 materials-15-03100-f007:**
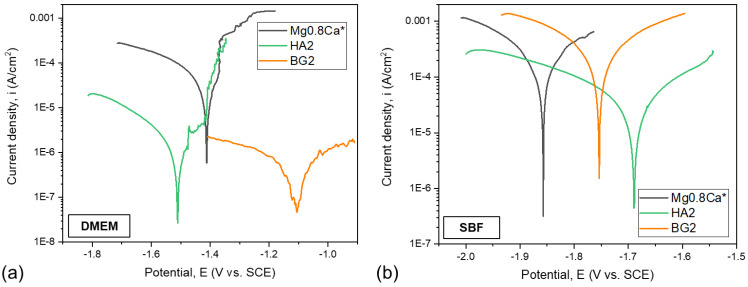
Tafel plots of the bare and HA2- and BG2-coated Mg-0.8Ca specimens immersed in (**a**) DMEM and (**b**) SBF testing media. * Mg-0.8Ca Tafel plots inserted from [41].

**Figure 8 materials-15-03100-f008:**
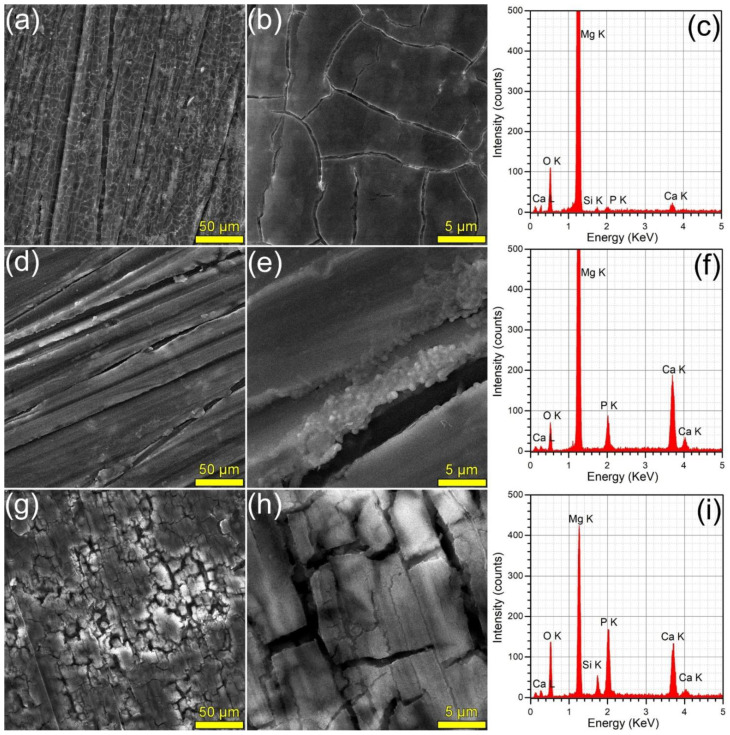
(**a**,**b**,**d**,**e**,**g**,**h**) SEM images and (**c**,**f**,**i**) EDS spectra collected subsequent to the corrosion tests performed in DMEM, on the (**a**–**c**) bare and (**d**–**f**) HA2- and (**g**–**i**) BG2-coated Mg-0.8Ca specimens.

**Figure 9 materials-15-03100-f009:**
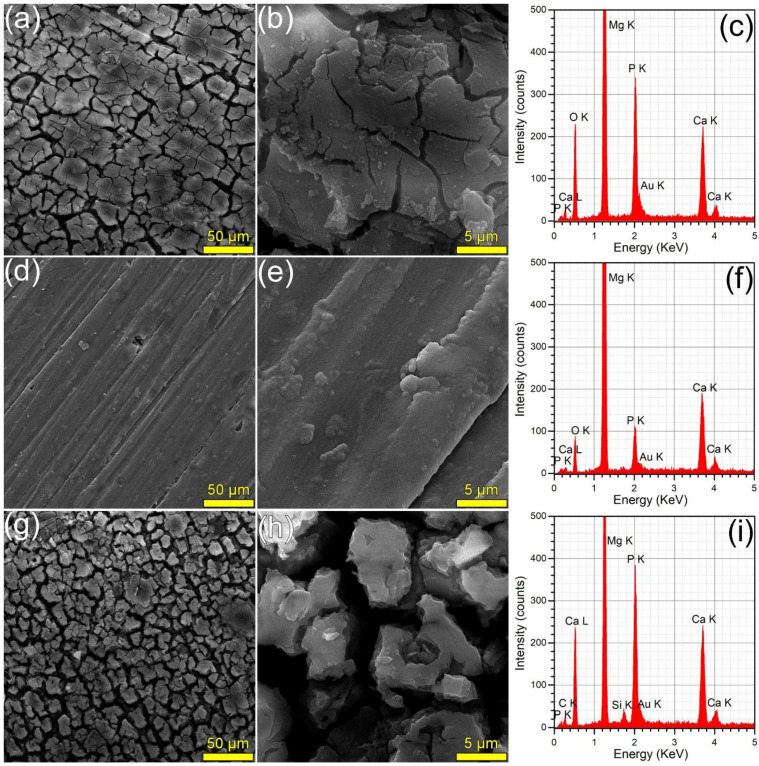
(**a**,**b**,**d**,**e**,**g**,**h**) SEM images and (**c**,**f**,**i**) EDS spectra collected subsequent to the corrosion tests performed in SBF, on the (**a**–**c**) bare and (**d**–**f**) HA2- and (**g**–**i**) BG2-coated Mg-0.8Ca specimens.

**Figure 10 materials-15-03100-f010:**
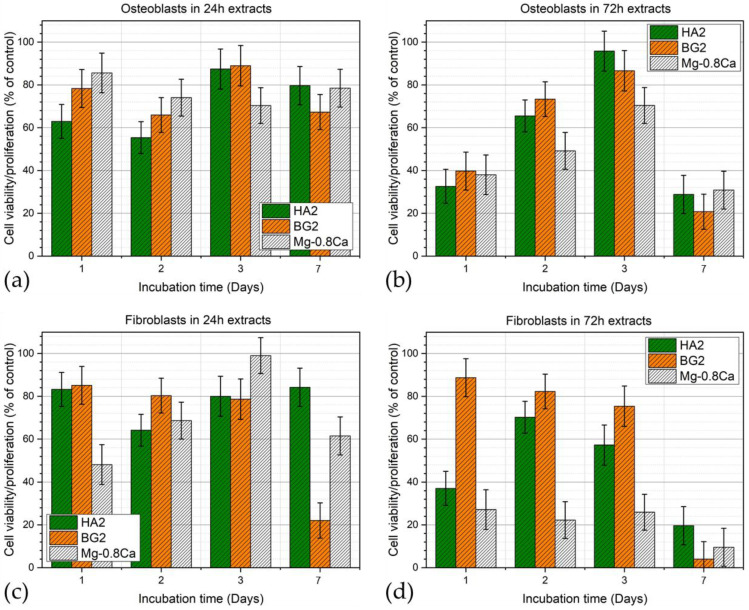
Viability/proliferation of (**a**,**b**) osteoblast and (**c**,**d**) fibroblast cells cultured for 1, 2, 3, and 7 days in (**a**,**c**) 24 h and (**b**,**d**) 72 h extracts obtained for the bare and HA2- and BG2-coated Mg-0.8Ca specimens.

**Table 1 materials-15-03100-t001:** Composition of the SBF and DMEM solutions.

Type of Medium	SBF (ISO 23317:2014)	Dmem
	*Ion concentration (mM)*	
Na^+^	142	156
K^+^	5	5.33
Mg^2+^	1.5	0.81
Ca^2+^	2.5	1.8
Cl^−^	147.8	121
HCO_3_^−^	4.2	0.91
HPO_4_^2−^	1.0	44.1
SO_4_^2−^	0.5	0.81
Ca/P molar ratio	2.5	1.98
	*pH buffer*	
	Tris-hydroxymethyl aminomethane	HEPES, [4-(2-hydroxyethyl)-1-piperazineethanesulfonic acid]
	*Organic components (mM)*	
AMINO ACIDS		
Glycine	-	0.4
L-Arginine hydrochloride	-	0.3981
L-Cystine 2HCl	-	0.2013
L-Glutamine	-	4.0
L-Histidine hydrochloride-H_2_O	-	0.2
L-Isoleucine	-	0.8015
L-Leucine	-	0.8015
L-Lysine hydrochloride	-	0.7978
L-Methionine	-	0.2013
L-Phenylalanine	-	0.4
L-Serine	-	0.4
L-Threonine	-	0.7983
L-Tryptophan	-	0.0784
L-Tyrosine disodium salt dihydrate	-	0.3985
L-Valine	-	0.8034
VITAMINS		
Choline chloride	-	0.0286
D-Calcium pantothenate	-	0.0084
Folic acid	-	0.0091
Niacinamide	-	0.0328
Pyridoxine hydrochloride	-	0.0196
Riboflavin	-	0.0011
Thiamine hydrochloride	-	0.0119
i-Inositol	-	0.04
OTHER		
D-Glucose	-	5.5–25
Sodium Pyruvate	-	0.0399
Phenol red	-	1.0

**Table 2 materials-15-03100-t002:** The main parameters of the electrochemical corrosion process in DMEM.

Sample	E_oc_ (V)	E_cor_ (V)	i_cor_ (µA/cm^2^)	β_c_ (mV)	β_a_ (mV)	R_p_ (kΩ × cm^2^)	P_e_ (%)
Mg-0.8Ca	−1.498	−1.411	50.8	280.3	165.8	0.89	-
HA2	−1.564	−1.51	2.0	270.3	161.4	21.2	95.92
BG2	−1.12	−1.10	0.8	686.2	297.0	103.20	98.28

**Table 3 materials-15-03100-t003:** The main parameters of the electrochemical corrosion process in SBF.

Sample	E_oc_ (V)	E_cor_ (V)	i_cor_ (µA/cm^2^)	β_c_ (mV)	β_a_ (mV)	R_p_ (kΩ × cm^2^)	P_e_ (%)
Mg-0.8Ca	−1.86	−1.85	576.0	404.3	350.5	0.14	-
HA2	−1.75	−1.68	62.2	453.2	282.9	1.21	89.19
BG2	−1.77	−1.75	1232.0	556.3	541.3	0.09	negative

## Data Availability

The raw data can be made available upon request addressed to the corresponding author.

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
