# Peer review of "Electrochemical and In Vitro Biological Evaluation of Bio-Active Coatings Deposited by Magnetron Sputtering onto Biocompatible Mg-0.8Ca Alloy"

_materials, 2022, doi:10.3390/ma15093100_

Round 1

Reviewer 1 Report

In this work, the crystalline carbonated hydroxyapatite (HA) and silica-rich glass (BG) layers were deposited by magnetron sputtering onto biodegradable Mg-0.8Ca alloys. The structural, surface energy, corrosion (in both mainly inorganic – SBF, and complex organic-inorganic – Dulbecco's Modified Eagle Medium – simulated body media) and cytocompatibility properties of hydroxyapatite and silica-rich thin layers deposited by RF-MS onto a Mg-0.8Ca-type (0.8 wt.% of Ca) biodegradable magnesium alloy were investigated. The work of the paper is well-done, with some interesting results. However, it requires an appropriate revision.

  1. More experimental details about of fabrication of the Mg-0.8Ca alloy and deposition of coatings by RF-MS should be provided.
  2. Micro and micro images of the coatings should be provided.
  3. More theoretical analyzes about the difference in electrochemical and in-vitro biological evaluation of bio-active coatings deposited by magnetron sputtering onto biocompatible Mg-Ca alloys are welcome.
  4. A proper improvement in English is needed.

Reviewer 2 Report

Review of paper no. materials-1682425 titled Comparative electrochemical and in-vitro biological evaluation of bio-active coatings deposited by magnetron sputtering onto biocompatible Mg-Ca alloys by A.-I. Bita et al.

This is an interesting and well-researched paper that studied the corrosion resistance and biological compatibility of a coated Mg-0.8Ca alloy. The alloy was coated with hydroxyapatite (HAP) and silica-rich glass (SRG). The HAP coating was found to have a better corrosion resistance. The paper is written in perfect English. It is acceptable for publication subject to revision.

1.The paper studied only one alloy. As such, the title should be shortened and changed to the following: Electrochemical and biological evaluation of bio-active coatings deposited by magnetron sputtering on a biocompatible Mg-0.8Ca alloy.

2.The choice of the coatings is not sufficiently explained. It is necessary to highlight the mechanical and biological properties of HAP in the introduction (https://doi.org/10.1016/j.actbio.2007.10.006, https://doi.org/10.1039/C6RA11679E).

3.What was the thickness of the coatings? It would be helpful to see the cross sections.

4.The corrosion resistance was studied in a simulated body fluid (SBF) and Dulbecco’s modified eagle medium (DMEM). What are the main agents of SBF and DMEM? You should present the chemical composition of the solutions.

5.The FTIR spectra are usually presented with a transmittance on the y axis. It is quite unusual to see the FTIR spectra with the absorbance on the y axis (Fig. 3b).

6.The polarization plots (Fig. 7) should be given with the potential on the x-axis and the current density on the y-axis. The potential is the independent variable.

7.The chemical composition of the corrosion products (Figs. 8c, f, i) should be tabulated and presented in at.%. It is not enough to show the overall EDS spectra. The same applies to Fig. 9 c, f, and i. Furthermore, it might be helpful to explain the appearance of various corrosion products using chemical equations.

8.The cytocompatibility (Section 3.3.5) is a major highlight. It needs to be further discussed. Two sentences (lines 480 - 484) are insufficient. Furthermore, the plots in Figs. 10 and 11 should be normalized to 100 %. Cell viability cannot be higher than 100 %. Also, please, improve the color-coding of the figures.

Round 2

Reviewer 2 Report

Authors answered most of my comments. The current manuscript can be accepted for publication.